# Network potential identifies therapeutic *miRNA* cocktails in Ewing sarcoma

**Davis T. Weaver**[1,2], **Kathleen I. Pishas**[3], **Drew Williamson**[4], **Jessica Scarborough**[1,2], **Stephen L. Lessnick**[5], **Andrew Dhawan**[2,6]*, **Jacob G. Scott**[1,2,7]*

**1** Case Western Reserve University School of Medicine, Cleveland, Ohio, United States of America, **2** Translational Hematology Oncology Research, Cleveland Clinic, Cleveland, Ohio, United States of America, **3** Peter MacCallum Cancer Centre, Melbourne, Australia, **4** Department of Pathology, Brigham & Women's Hospital, Boston, Massachusetts, United States of America, **5** Nationwide Children's Hospital, Columbus, Ohio, United States of America, **6** Division of Neurology, Cleveland Clinic, Cleveland, Ohio, United States of America, **7** Department of Physics, Case Western Reserve University, Cleveland, Ohio, United States of America

* dhawana@ccf.org (AD); scottj10@ccf.org (JGS)

**Data Availability Statement:** All of the software we developed for this project can be found on Github. (https://github.com/DavisWeaver/MiR_Combo_Targeting and https://github.com/DavisWeaver/

## Abstract

MicroRNA (miRNA)-based therapies are an emerging class of targeted therapeutics with many potential applications. Ewing Sarcoma patients could benefit dramatically from personalized miRNA therapy due to inter-patient heterogeneity and a lack of druggable (to this point) targets. However, because of the broad effects miRNAs may have on different cells and tissues, trials of miRNA therapies have struggled due to severe toxicity and unanticipated immune response. In order to overcome this hurdle, a network science-based approach is well-equipped to evaluate and identify miRNA candidates and combinations of candidates for the repression of key oncogenic targets while avoiding repression of essential housekeeping genes. We first characterized 6 Ewing sarcoma cell lines using mRNA sequencing. We then estimated a measure of tumor state, which we term network potential, based on both the mRNA gene expression and the underlying protein-protein interaction network in the tumor. Next, we ranked mRNA targets based on their contribution to network potential. We then identified miRNAs and combinations of miRNAs that preferentially act to repress mRNA targets with the greatest influence on network potential. Our analysis identified TRIM25, APP, ELAV1, RNF4, and HNRNPL as ideal mRNA targets for Ewing sarcoma therapy. Using predicted miRNA-mRNA target mappings, we identified miR-3613-3p, let-7a-3p, miR-300, miR-424-5p, and let-7b-3p as candidate optimal miRNAs for preferential repression of these targets. Ultimately, our work, as exemplified in the case of Ewing sarcoma, describes a novel pipeline by which personalized miRNA cocktails can be designed to maximally perturb gene networks contributing to cancer progression.

## Author summary

Precision medicine in cancer aims to find the right treatment, for the right patient, at the right time. Substantial variation between patient tumors, even of the same disease site, has

disruptr). All processed data needed for reproduction of the results of the paper are available in the same repository. All raw data files were published on GEA at https://www.ncbi.nlm.nih.gov/geo/query/acc.cgi?acc=GSE98787 (accession number GSE98787).

**Funding:** JGS thanks the NIH for their support through NIH R37CA244613 and the Paul Calabresi Career Development Award for Clinical Oncology (NIH K12CA076917). KIP acknowledges financial support from the NHMRC CJ Martin Overseas Biomedical Fellowship (APP1111032) and Alex's Lemonade Stand Young Investigator Grant (APP37138). DTW acknowledges financial support from the Cancer Center Trainee Award for Cancer Research from the Case Comprehensive Cancer Center. JS acknowledges financial support from the NIH for their support through NIH T32GM007250. The funders had no role in study design, data collection and analysis, decision to publish, or preparation of the manuscript.

**Competing interests:** The authors have declared that no competing interests exist.

limited the application of precision medicine in the clinic. In this study, we present novel computational tools for the identification of targets for cancer therapy using widely available sequencing data. We used a network-science based approach that leveraged multiple types of 'omic data to identify functionally relevant disease targets. Further, we developed algorithms to identify potential miRNA-based therapies that inhibit these predicted disease targets. We applied this pipeline to a novel Ewing Sarcoma transcriptomics data-set as well as publicly available patient data from the St. Jude Cloud. We identified a number of promising therapeutic targets for this rare disease, including EWSR1, the proposed driver of Ewing Sarcoma development. These novel data and methods will provide researchers with new tools for the development of precision medicine treatments in a variety of cancer systems.

## Introduction

Ewing sarcoma is a rare malignancy arising from a gene fusion secondary to rearrangements involving the EWS gene [1]. There are 200–300 reported cases each year in the United States, disproportionately affecting children [2]. High levels of inter-tumor heterogeneity are observed among Ewing sarcoma patients despite a shared EWS gene fusion initiating event [3]. Ewing sarcoma is also extremely prone to developing resistance to available chemotherapeutics [4]. These features make it an ideal system to develop personalized therapies for resistant tumors or to avoid the development of resistance altogether.

MicroRNA (miRNA)-based therapeutics, including anti-sense oligonucleotides, are an emerging class of cancer therapy [5]. Recent work has highlighted the critical importance of miRNAs in the development and maintenance of the cancer phenotype [4–6]. MiRNA dysregulation has been implicated in the development of each of the hallmark features of cancer [7], and restoration of expression of some of these critical downregulated miRNAs has been studied as a potential treatment for several different cancers [6, 8, 9]. In particular, in the past decade, anti-sense oligonucleotide inhibitors of the STAT3 transcription factor have shown promise in the settings of lymphoma [10, 11] and neuroblastoma [12]. MiR-34 has shown to be effective in pre-clinical studies for treatment of both lung cancer [13–15] and prostate cancer [16]. Finally, miR-34 and let-7 combination therapy has been shown to be effective in pre-clinical studies of lung cancer [15].

MiRNAs have been recognized as potential high-value therapeutics in part due to their ability to cause widespread changes in a cell-signaling network [5]. A single miRNA molecule can bind to and repress multiple mRNA transcripts [6, 17–19], a property that can be exploited when designing therapy to maximally disrupt a cancer cell signaling network.

This promiscuity of miRNA binding may also increase the risk of off-target effects and toxicity (Fig 1). For example, miR-34 was effective in pre-clinical studies for the treatment of a variety of solid tumors [13–16], only to fail in a phase I clinical trial due to "immune-related serious adverse events" [20]. To capitalize on the promise of miRNA-based cancer therapy while limiting potential toxicity, we developed a systematic, network-based approach to evaluate miRNA cocktails. We focused on miRNA cocktails rather than single miRNA therapeutics due to the potential for miRNA cocktails to minimize toxicity compared to single miRNA regimens [21] (Fig 1).

In this work, we build on previous studies applying thermodynamic measures to cell signaling networks in the field of cancer biology [22–24], as well as works that describe a method to use gene homology to map miRNAs to the mRNA transcripts they likely repress [6, 17, 18].

Reitman et al. previously described a metric of cell state analogous to Gibbs free energy that can be calculated using the protein-protein interaction network of human cells and corresponding transcriptomic data [22]. Gibbs free energy has been correlated with a number of cancer-specific outcomes, including cancer grade and patient survival [23]. Additionally, Reitman et al. leveraged Gibbs and other network measures to identify personalized protein targets for therapy in a dataset of low-grade glioma patients from The Cancer Genome Atlas (TCGA) [22]. Previous work has also highlighted the critical importance of miRNAs to maintenance and development of the oncogenic phenotype, and demonstrated the utility of applying miRNA-mRNA mappings. [6] In this work, we developed and applied a computational pipeline that leverages these network principles to identify miRNA cocktails for the treatment of Ewing sarcoma.

## Materials and methods

### 0.1 Overview

We characterized six previously described Ewing sarcoma cell lines in triplicate [25]—A673, ES2, EWS502, TC252, TC32, and TC71—using paired miRNA and mRNA sequencing. By evaluating 6 distinct cell lines, we aimed to assess the heterogeneity inherent to Ewing sarcoma *in-vitro*. We also utilized mRNA sequencing data for 15 ewing sarcoma patient tumor samples made available on the St. Jude Cloud [26]. We then defined a measure of tumor state, which we term network potential (Eq 1), based on both mRNA gene expression and the underlying protein-protein interaction (PPI) network. Next, we ranked mRNA targets based on their contribution to network potential of each cell line, aiming to approximate the relative importance of each mRNA to network stability. Relative importance of each mRNA to network stability was determined by calculating the change in network potential of each network before and after *in silico* repression of each mRNA ($\Delta G$, described in **Section 0.5**). After identifying these mRNA targets, we then identified miRNA and miRNA cocktails that preferentially acted to repress the most influential of the ranked mRNA targets, with the aim of defining synthetic miRNA-based therapy for down-regulation of these targets. Our computational pipeline is schematized in Fig 2.

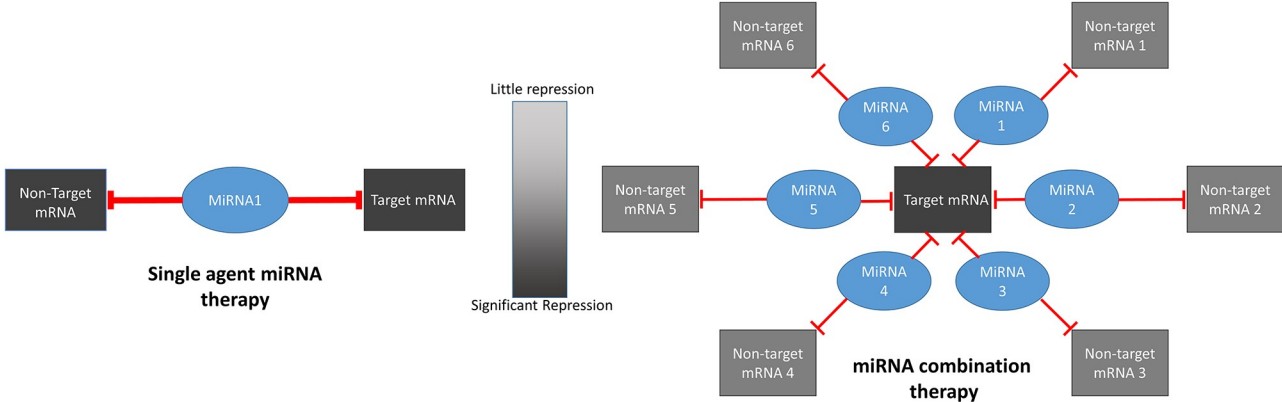

**Fig 1. Cartoon describing rationale for focusing on miRNA combination therapy.** With single-agent therapy, both target mRNA and non-target mRNA are inhibited an equal amount, potentially resulting in toxicity due to off-target effects. With miRNA combination therapy, the common target mRNA is inhibited to a greater degree than any individual non-target miRNA.

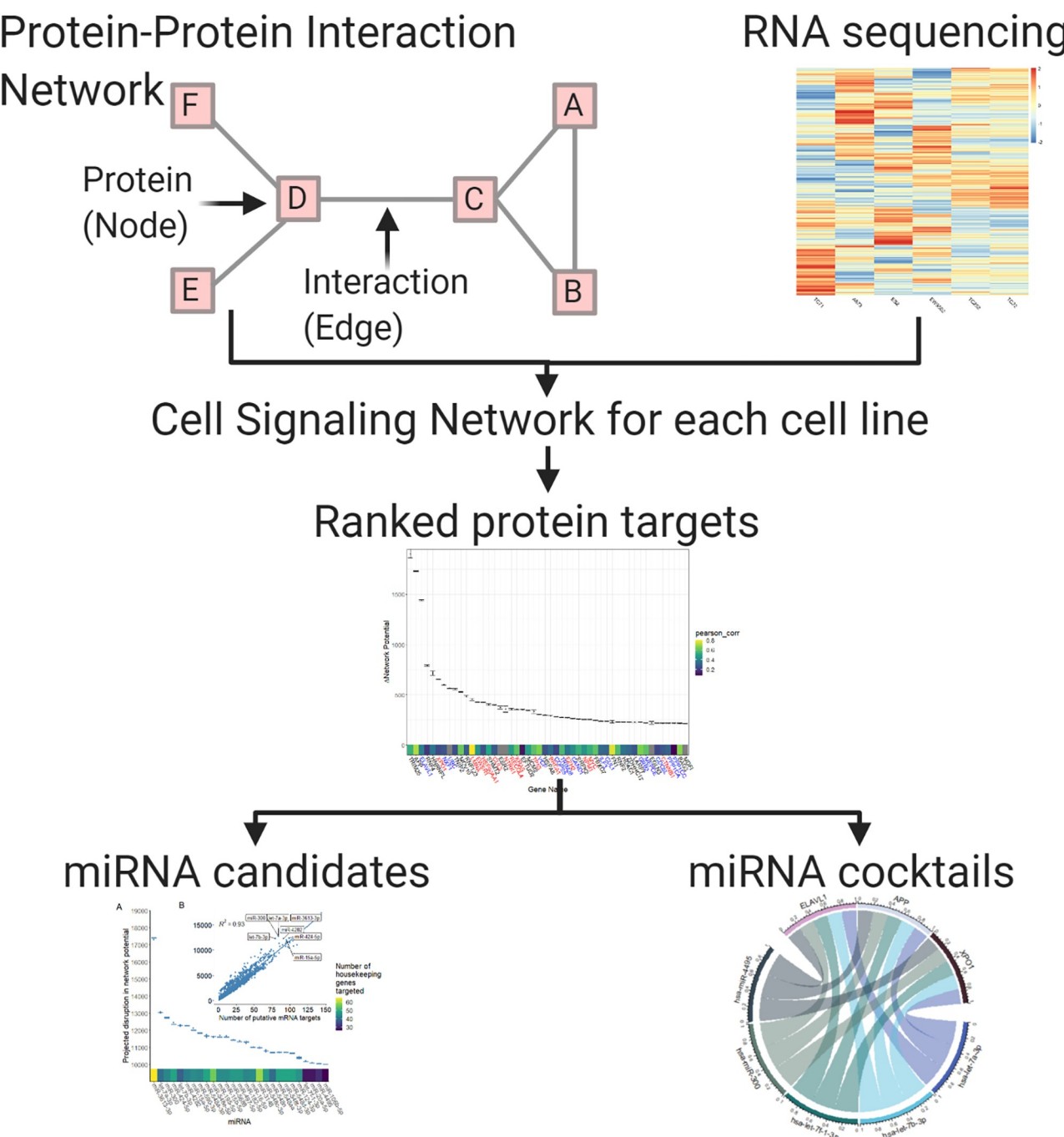

**Fig 2. Simplified schematic of our computational pipeline.** We defined a measure of tumor state, which we term network potential (Eq 1), based on both mRNA gene expression and the underlying protein-protein interaction (PPI) network. Next, we ranked mRNA targets based on their contribution to network potential of each cell line, aiming to approximate the relative importance of each mRNA to network stability. After identifying these mRNA targets, we then identified miRNA and miRNA cocktails that preferentially acted to repress the most influential of the ranked mRNA targets, with the aim of defining synthetic miRNA-based therapy for down-regulation of these targets.

## 0.2 Data sources

We utilized three data sources to develop our Ewing sarcoma cell signaling networks: the Bio-GRID protein-protein interaction database [27], mRNA expression data from 6 Ewing sarcoma cell lines, which are available on GEO (accession GSE98787), and mRNA expression data from 15 Ewing sarcoma patient samples, which are available on the St. Jude Cloud [26].

**Protein protein interaction databases.**   The BioGRID interaction database contains curated data detailing known interactions between proteins for a variety of different species, including Homo sapiens. The data were generated by manual curation of the biomedical literature to identify documented interactions between proteins [27]. To assist in manual curation, the BioGRID project uses a natural language processing algorithm that analyzes the scientific literature to identify manuscripts likely to contain information about novel PPIs. The dataset is therefore limited to protein interactions that are reliably reported in the scientific literature. As new research accumulates, substantial changes to the PPI network may occur. For example, between 2016 and 2018, the number of documented PPIs in *Homo sapiens* grew from 365,547 to 449,842. The 449,842 documented interactions in 2018 were identified through curation of 27,631 publications [27]. Importantly, the PPI network is designed to represent normal human tissue. To assess the importance of the specific PPI used to our results, we repeated much of our analysis using stringdb, another publicly available PPI with millions of documented interactions. To maintain some consistency with biogrid, we modulated the provided "interaction score" until the resulting network had around the same number of edges as the biogrid network. Interaction score is a number ranging from 0 to 1000 that describes the likelihood that a given protein protein interaction is biologically relevant. For our analysis, we included only interactions with an interaction score $\geq 700$.

**Ewing sarcoma transcriptomics.**   Second, we utilized mRNA expression data from *in vitro* experiments conducted on six Ewing sarcoma cell lines (3 biological replicates per cell line). RNA/miRNA extraction was performed with a Qiagen kit with on-column DNase digestion. These mRNA and miRNA expression data were then normalized to account for between sample differences in data processing and further adjusted using a regularized log (Rlog) transformation [28, 29]. In order to extend our study to patient samples, we repeated our analysis on 15 patient tumors from the St. Jude Cloud for which RNA sequencing data was available. The St. Jude Cloud is a comprehensive, cloud-based data-sharing ecosystem that provides genomic data on thousands of samples from patients with pediatric cancer [26].

Notably, methods for calculating network potential from this type of data require protein concentrations rather than mRNA transcript concentrations. For the purposes of this analysis, we assumed that concentration of protein in an Ewing sarcoma tumor was equivalent to the concentration of the relevant mRNA transcript. A large body of work suggests that mRNA levels are the primary driver of protein levels in a cell under steady state conditions (i.e. not undergoing proliferation, response to stress, differentiation etc) [30–33]. However, recent work in a 375 cancer cell lines has shown that mRNA expression may not be predictive of protein expression in the setting of malignancy [34]. For this reason, we included the protein-mRNA correlations from their experiments alongside some of our key findings to provide needed context.

## 0.3 Network development

We first developed a generic network to represent human cell signaling networks using the BioGRID interaction database [27]. The BioGRID protein-protein interaction network can be downloaded as a non-linear data structure containing ordered pairs of proteins and all the other proteins with which they interact. This data structure can be represented as an

undirected graph, with vertex set $\mathcal{V}$, where each vertex represents a protein, and edge set ($\mathcal{E}$) describes the interactions between proteins.

Using mRNA sequencing data from 6 Ewing sarcoma cell lines in triplicate, we then ascribed mRNA transcript concentration for each gene as an attribute to represent the protein concentration for each node in the graph. Through this process, we developed networks specific to each cell line and replicate in our study (18 total samples). We then repeated the same process to develop networks specifics to the 15 patient tumor samples.

## 0.4 Network potential calculation

Using the cell signaling network with attached cell line and replicate number specific normalized mRNA expression data, we defined a measure of tumor state following Reitman et al. [22], which we term network potential. Our Network potential metric was inspired by Gibbs free energy in physics or chemistry. We first calculate the network potential of the $i$-th node in the graph:

$$G_i = C_i \ln \left[ \frac{C_i}{\sum C_j + C_i} \right]. \tag{1}$$

where $G_i$ is equal to the network potential of an individual node of the graph, $C_i$ is equal to the concentration of protein corresponding to node $G_i$, and $C_j$ is the concentration of protein of the $j$-th neighbor of $G_i$. Because of the natural log transformation, $G_i$ will always return a negative number. Total network potential ($G$) of the network can then be calculated as the sum over all nodes:

$$G = \sum_i G_i. \tag{2}$$

where $G$ is equal to the total network potential for each biological replicate of a given cell line. We then compared total network potential across cell lines and biological replicates. More negative network potentials were interpreted as being "larger" in the absolute sense.

## 0.5 Ranking of protein targets

After calculating network potential for each node and the full network, we simulated "repression" of every node in each network by reducing their expression (computationally) to zero, individually [35]. Clinically, this would be akin to the application of a drug that perfectly inhibited the protein/mRNA of interest. Next, we re-calculated network potential for the full network and calculated the change in network potential ($\Delta G$) by subtracting the new network potential value for the network potential value of the "unrepressed" network. Because the network potential of each node was negative, systematic repression of a given node always drove the total network potential to be less negative. As a result, this approach will always return a positive $\Delta G$. We then ranked each node in the network according to $\Delta G$ for further analysis.

We also evaluated the top predicted genes by $\Delta G$ against a null model of $\Delta G$ to evaluate the likelihood that these observed disruptions were due to random chance. To construct our null distribution of $\Delta G$, we repeated the following process 1000 times for each sample under study:

1. We constructed a random graph that preserves the original degree distribution for the underlying protein-protein interaction network by iteratively swapping edges. For each random graph, we performed n*100 swaps where n is the number of nodes in the original graph.

2. We then constructed a cell signaling network using the random graph and the mRNA expression data for that sample. mRNA expression was left unchanged for each random graph.

3. We computed the $\Delta G$ for the top 50 proteins under study on the new random graph.

We then calculated the average and standard deviation $\Delta G$ from all 1000 iterations of the above process to compute a bootstrapped null distribution of $\Delta G$. We then computed confidence intervals for $\Delta G$, employing the bonferroni correction to account for multiple hypothesis testing.

Our pipeline was designed to make use of parallel computing on the high-performance cluster (HPC) at Case Western Reserve University.

## 0.6 Identification of miRNA cocktails

To generate miRNA-mRNA mappings, we implemented a protocol described previously [36]. Briefly, we identified all predicted mRNA targets for each miRNA in our dataset using the miRNAtap database in R, version 1.18.0, as implemented through the Bioconductor targetscan org.Hs.eg.db package, version 3.8.2 [17]. We used all five possible databases (default settings): DIANA version 5.061 [19], Miranda 2010 release62 [37], PicTar 2005 release63 [38], TargetScan 7.164 [39] and miRDB 5.065 [18], with a minimum source number of 2, and the union of all targets found was taken as the set of targets for a given miRNA. Through this mapping, we identified a list of mRNA transcripts that are predicted to be repressed by a given miRNA. Our code and processed data files are available on Github at: https://github.com/DavisWeaver/MiR_Combo_Targeting/.

Using this mapping, as well as our ranked list of promising gene candidates for repression from our network analysis, we were able to identify a list of miRNA that we predict would maximally disrupt the Ewing sarcoma cell signaling network when introduced synthetically. To rank miRNA targets, we first identified all the genes on the full target list that a given miRNA was predicted to repress (described in Section 0.5). Next, we summed the predicted $\Delta G$ when each of these genes was repressed *in silico* to generate the maximum potential disruption that could be achieved if a given miRNA were introduced synthetically into an Ewing sarcoma tumor. We then ranked miRNA candidates in descending order of the maximum predicted network disruption.

Given the documented cases of systemic toxicities associated with miRNA-based therapies, the miRNA that inhibits the most targets might not necessarily be the best drug target. We therefore sought to identify combinations of miRNAs that individually repressed key drug targets, while avoiding repression of housekeeping genes that may lead to toxicity. We defined housekeeping genes using a previously described gene set [40]. In this study, housekeeping genes were identified by evaluating RNA sequencing data from a large number of normal tissue samples. Genes that are consistently expressed in all or nearly all tissue types were assumed to be so-called housekeeping genes. Our hypothesis is that by giving a cocktail of miRNAs with predicted activity against one or multiple identified drug targets, each individual miRNA could be given at a low dose such that only the mRNA transcripts that are targeted by multiple miRNAs in the cocktail are affected (Fig 1). Also with an eye towards limiting toxicity, we restrained our search to endogenous miRNAs rather than broadening to engineered exogenous miRNA mimics. To that end, we designed a loss function (see equation below) to balance

the the effects of repressing the housekeeping gene set $I$ as well as the target gene set $J$:

$$L(\mu) = \sum_{i,j} A(c)(G_i) - A(c)(G_j) \quad \text{for } i \in I \text{ and } j \in J, \tag{3}$$

$$A(c) = \begin{cases} 0, & \text{if } c \leq 1 \\ 0.2c, & \text{if } 1 < c < 6 \\ 1, & \text{otherwise} \end{cases} \tag{4}$$

Where $A(c)$ determines the degree of repression as a function of the number of times, $c$, that a given gene, $i$, is targeted by a given miRNA cocktail, $\mu$. We recognize that assuming each additional miRNA additively represses 20% of a given gene is somewhat arbitrary. Future work could improve on this miRNA cocktail optimization approach by more formally addressing miRNA repression in different contexts.

We first transformed the projected change in network potential for each gene such that housekeeping genes exerted a positive change in network potential and the top 10 predicted targets exerted a negative change in network potential. We then ranked 3-miRNA combinations according to their projected effect on network potential, where more negative changes in network potential were interpreted as most effective for maximizing on-target effects while minimizing off-target effects. As a further constraint, a gene had to be targeted by 2 or more miRNA in a given cocktail to be considered repressed. Each miRNA was assumed to downregulate a given gene by 20%, such that genes targeted by 2 miRs were assumed to have their expression decreased by 40%, and genes targeted by 3 miRs were assumed to have their expression decreased by 60%. We repeated our analysis, varying between 10% and 50% repression to assess the impact of this assumption on our predicted miRNA cocktails. Rather than evaluate every potential 3-miRNA combination, we limited our analysis to miRNA that target at least 2 of our 10 target genes. These constraints were defined *a priori*. We repeated this analysis to identify cocktails that target larger or smaller groups of mRNA (the top 5 or 15 mRNA targets) in order to assess the stability of the predicted cocktail to changing conditions.

## Results

### 0.7 Network overview

We calculated the network potential, a unitless measure of cell state, for each protein in the cell signaling networks for each of the six Ewing sarcoma cell lines in our experiment. An overview comparing network potential to normalized mRNA expression can be found in Fig 3. An additional overview of the total network potential for each cell line and biological replicate compared to total mRNA expression is presented in S1 Fig.

The histograms of network potential and mRNA expression demonstrate markedly different distributions (S1 Fig), indicating that network potential describes different features of a cell signaling network compared to mRNA expression alone. Notably, network potential and mRNA expression for these cell lines are stable across different biological replicates, as demonstrated by the low interquartile range (S1C and S1D Fig). There were larger differences in both mean expression and network potential across cell lines (S1C and S1D Fig when compared to between-replicate differences. The global average network potential across all samples was $-3.4 \times 10^5$ with a standard deviation of 1605.

The included patient samples (beginning with SJEWS on S1 Fig) demonstrate substantially more variation in both mRNA expression and network potential. Much of what we are

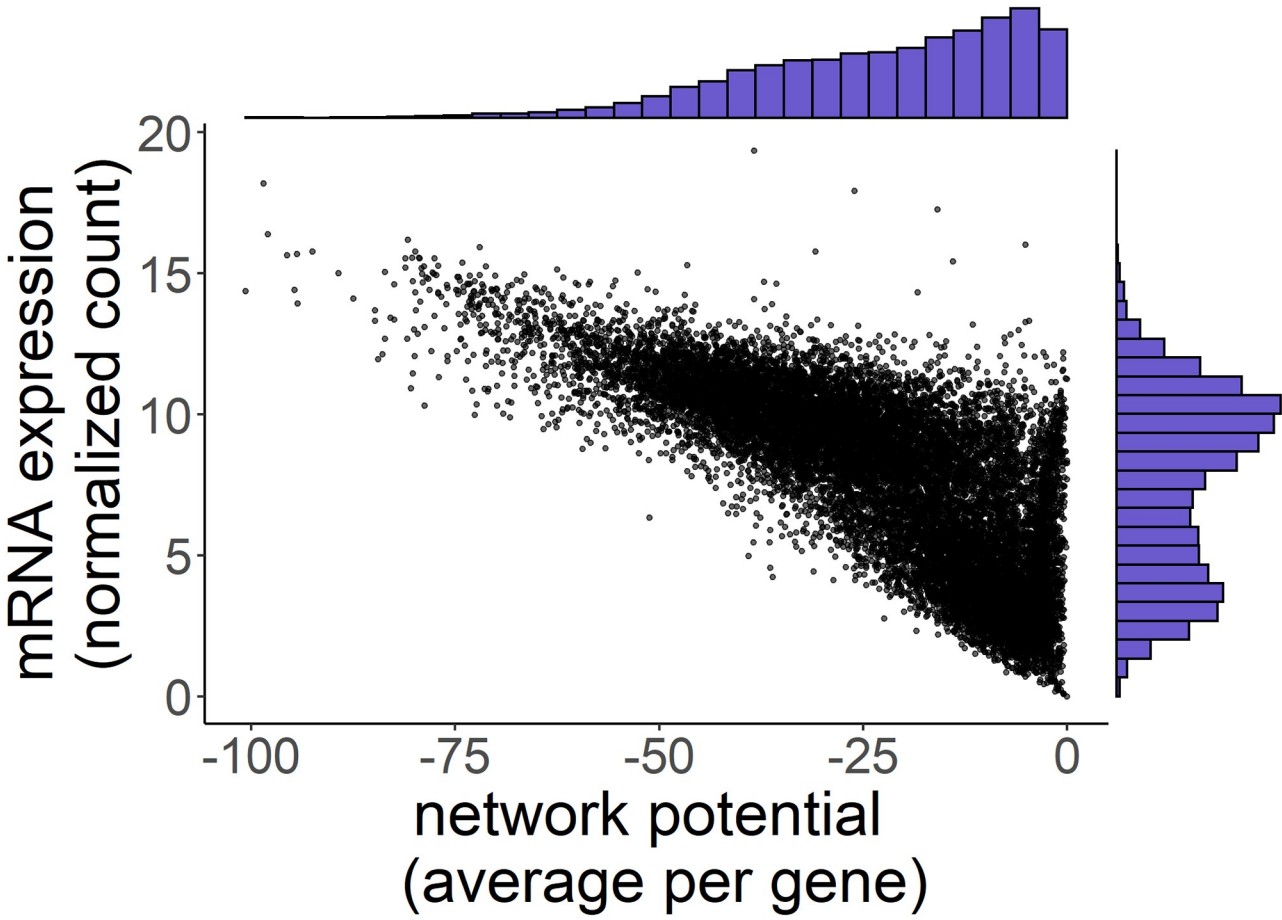

**Fig 3. Network potential demonstrates a different distribution compared to mRNA expression.** Main panel: scatterplot comparing mRNA expression and network potential for all genes in our 18 Ewing Sarcoma cell lines. For each gene, we averaged across all samples for both mRNA expression and network potential. Unlike network potential (top axis histogram), mRNA expression (right axis histogram) has a bimodal distribution.

capturing here can likely be attributed to batch effects, as these patient samples may have been sequenced years apart on machines with dramatically different capabilities.

## 0.8 Identification of protein targets

We identified TRIM25, APP, ELAV1, RNF4, and XPO1 as top 5 targets for therapy for each of the 6 cell lines based on the degree of network disruption induced following *in silico* repression of each gene. Of these 5 genes, only XP01 has been previously implicated in oncogenesis [41], while only ELAVL1 is a known essential housekeeping gene. There was a high degree of concordance between cell lines among the top predicted targets (S1 Table). Of the top ten predicted targets, all 10 targets are conserved for all 6 cell lines. The top 50 protein targets are presented in Fig 4. The top 50 protein targets, limited to those causally implicated in cancer, can be found in S2 Fig. Many of the top identified genes fell within the 99.99% confidence interval of the computed null distribution (Fig 4), suggesting that these genes are highly connected hub genes that are likely to score high in $\Delta G$ regardless of the tumor-specific RNA-sequencing information provided. As a sensitivity analysis, we repeated our protein identification pipeline using stringdb as the PPI network provider rather than biogrid. Using stringdb, we identified broadly similar targets, with the majority of identified genes being either essential

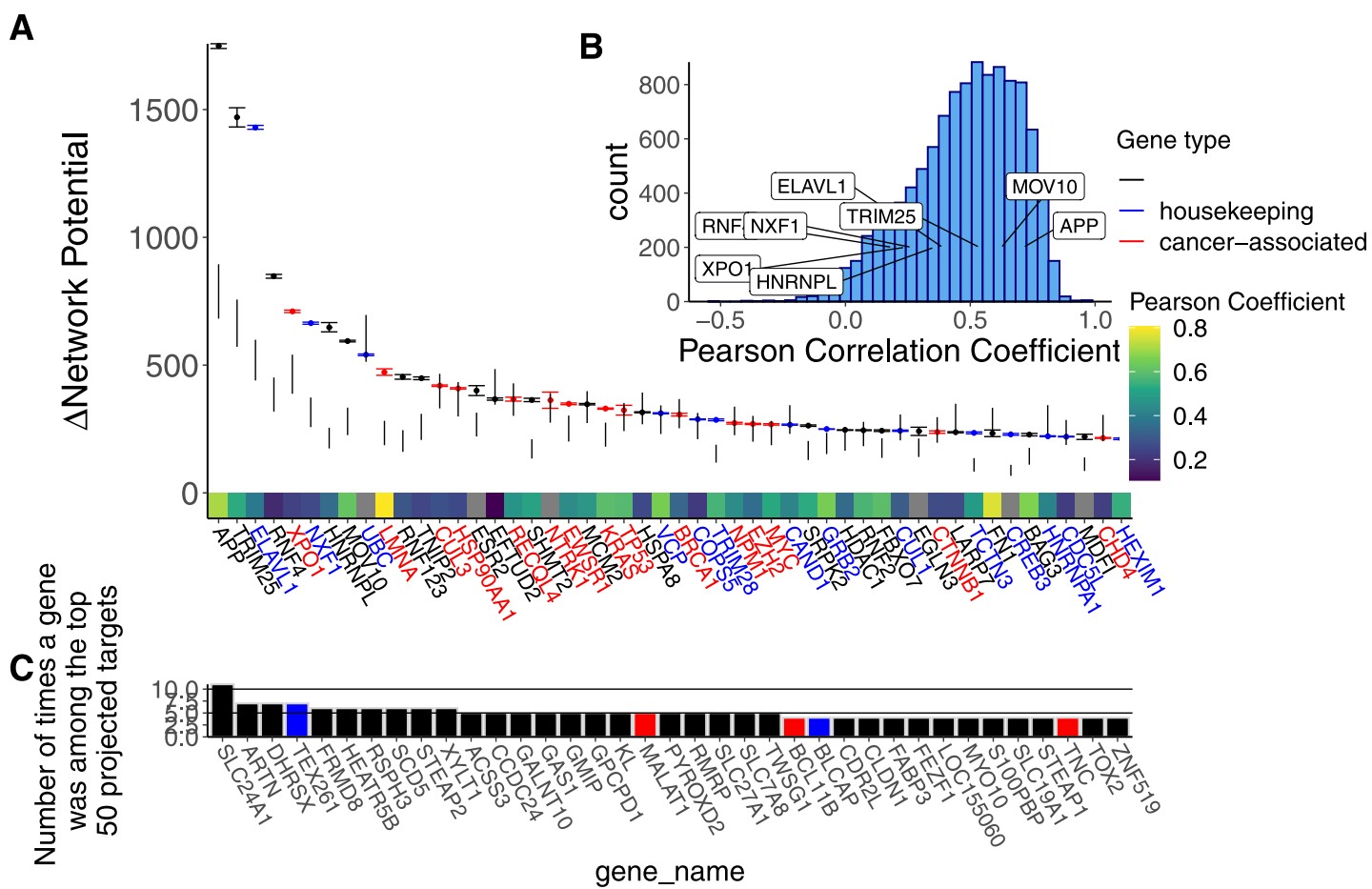

**Fig 4. TRIM25, APP, ELAVL1, AND RNF4, and XPO1 are the top protein targets ranked by predicted disruption following *in silico* repression. Panel A**: Box and whisker plot describing the change in network potential following *in silico* repression for each of the top 50 proteins. 99.99% confidence interval from the permutation test are displayed alongside the box and whisker plots. It is notable that EWSR1, the kinase associated with Ewing sarcoma development, is considered highly influential in the cell signaling network by this method, even in comparison to the computed null distribution. Genes that have previously been causally implicated in cancer according to the Cosmic database are highlighted in red [41]. Essential housekeeping genes (excluding those that are causally implicated in cancer) are highlighted in blue. The heat-map on the x-axis corresponds to the protein-mRNA correlation of each gene in the Cancer Cell Line Encyclopedia [34]. **Panel B**: Histogram depicting the distribution of Pearson correlation between mRNA expression and protein expression from the Cancer Cell Line Encyclopedia for all nodes included in our final Ewing sarcoma cell signaling networks. Proteins that were ranked particularly highly in panel A were labeled in panel B. **Pancel C**: Bar chart describing the most frequently observed genes among the top projected targets for the 15 patient tumor samples we analyzed. There was very little overlap in top projected targets between the cell line and patient data, reflecting the transcriptional heterogeneity present in Ewing Sarcoma.

housekeeping genes or genes causally implicated in cancer. 13 out of the 50 identified targets were shared between stringdb and biogrid. In contrast to our main results using biogrid, the vast majority of top targets identified with stringdb fell within the 99.99% confidence interval of the computed null distribution (S3 Fig) In addition, the distribution of network potential across all genes looks extremely similar regardless of which PPI was used (Fig 3 and S3 Fig).

Surprisingly, there was very little overlap between the top predicted targets between the cell line data and the patient samples (Fig 4). The genes that appeared most frequently among the top projected targets for the 15 patient tumor samples were SLC24A1, ARTN, DHRSX, TEX261, and FRMD8. Compared to the cell line samples, fewer of the most frequent projected targets were cancer-associated or defined housekeeping genes (Fig 4).

Some of these identified genes in the cell line data are likely essential housekeeping genes highly expressed in all or most cells in the body, making them inappropriate drug targets (Fig

[4]). TRIM25, and ELAV1, for example, are involved in protein modification and RNA binding, respectively [42]. We therefore repeated this analysis, limiting our search to gene targets that have been causally implicated in cancer [41]. With this limitation in place, we identified XPO1, LMNA, EWSR1, HSP90AA1, and CUL3 as the top 5 targets for therapy when $\Delta G$ was averaged for all cell lines. The top 10 cancer-related targets for each cell line can be found in (S1 Table).

We also conducted gene set enrichment analysis for the all the genes represented in our cell signaling network (averaged across all samples). We ranked genes by network potential (averaged across all samples) and compared our gene set to the "hallmarks" pathways set, downloaded from the Molecular Signatures Database (MSigDB) [43, 44]. This analysis was conducted using the fGSEA package in R, which uses the Benjamini—Hochberg procedure to correct the false discovery rate [45, 46]. Our gene set was enriched (adjusted p-value < 0.05) in 24 of the 50 pathways included in the hallmarks set; including apoptosis, DNA repair, mTOR signaling, MYC signaling, and WNT $\beta$-catenin signaling. Our gene set was also highly enriched (normalized enrichment score = 1.73) in the miRNA bio-genesis pathway. The full results are presented in S3 Table.

## 0.9 Identification of miRNA cocktails

We identified several miRNAs that were predicted to dramatically disrupt the Ewing sarcoma cell signaling network (Fig 5). When averaging all cell lines, we identified miR-3613–3p, let-7a-3p, miR-300, miR-424–5p, and let-7b-3p as the ideal miRs for preferential repression of proteins predicted to be important for Ewing sarcoma signaling network stability. miR-3613–3p, let-7a-3p, miR-300, miR-424–5p, and let-7b-3p were predicted to cause an average network network potential increase (driving the system less negative) of 17382, 13034, 12746, 12364 and 12280, respectively (see Fig 6). It should also be noted that we were able to identify a substantial number of miRNAs with potential activity against the Ewing sarcoma cell signaling network. We identified 27 miRNAs with an average predicted network potential disruption of greater than 10, 000. For comparison, the largest network change in network potential that could be achieved with a single gene repression across all cell lines was just 2064 (TRIM25). miRNA sequencing of the 6 cell lines under study did not reveal any clear pattern of miRNA expression based on predicted network potential disruption.

These individual miRNAs target large numbers of transcripts in the cell and therefore may be difficult to administer as single-agents due to extreme toxicity. For example, the top miR candidate, miR-3613–3p, was predicted to repress 144 distinct mRNA transcripts in the full target set. We therefore sought to identify cocktails of miRNA that could cooperatively down-regulate key non-housekeeping genes while avoiding cooperative down-regulation of housekeeping genes that may be associated with toxicity. When targeting the top 10 predicted proteins from our *in silico* repression experiments, a 3 miRNA cocktail of miR-483–3p, miR-379–3p, and miR-345–5p was predicted to be the most optimal across all cell lines (Fig 6A and 6B). Under the same conditions, a 3-miR cocktail of miR-300, let-7b-3p, and let-7a-3p was predicted to be the least optimal among 16,215 tested combinations (Fig 6C and 6D). Notably, the most and least optimal miRNA combinations had similar activity against the 10 targets (Fig 6A and 6C). The worst cocktail was defined by high levels of cooperative downregulation of housekeeping genes rather than lack of efficacy against putative targets (Fig 6C and 6D). Let-7b-3p and let-7a-3p were heavily represented in the least optimal cocktails tested, appearing in 10 of the 10 worst 3 miRNA cocktails (Fig 6E). These highly promiscuous miRNA target large numbers of housekeeping genes, limiting their therapeutic utility alone or in combination (Fig 5B).

Notably, many of the most promising miRNA when considering only their total predicted network disruption tend to appear in the least optimal cocktails (Fig 5). This likely occurs

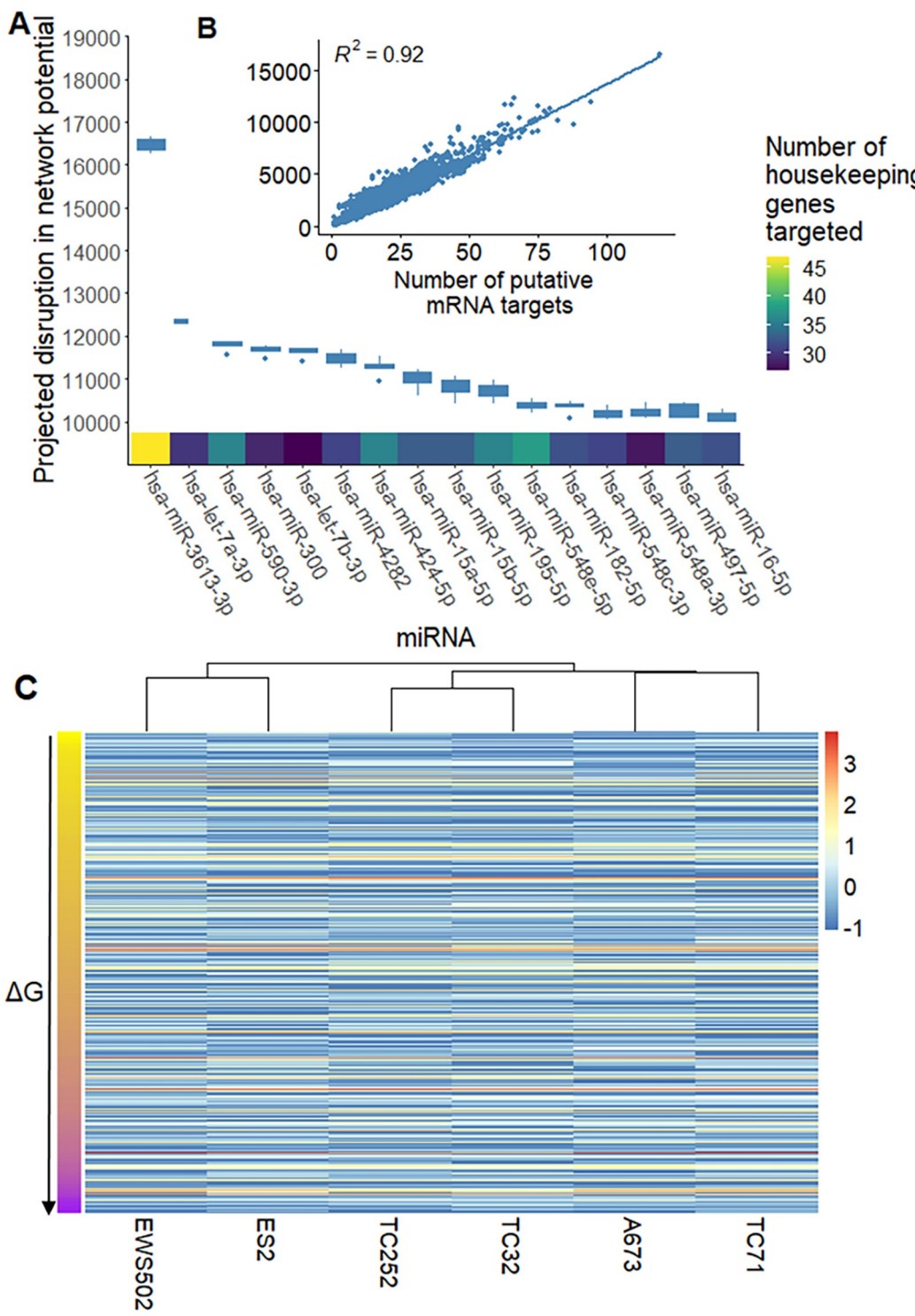

**Fig 5. Many of the most promising miRNA candidates repress large numbers of essential housekeeping genes.** We identified the top miRNA for treatment of Ewing sarcoma, ranked by their predicted disruption of the Ewing sarcoma cell signaling network. **A**: Boxplot showing the projected disruption in network potential for the top miRNA candidates (averaged across all samples). The heatmap on the x-axis describes the number of essential housekeeping genes that each miRNA is predicted to target. **B**: Scatterplot showing the relationship between projected network disruption and the number of putative mRNA targets for a given miRNA. **C**: Heatmap showing z-score normalized miRNA expression for 622 of the evaluated miRNA for the 6 cell lines under study. The Y axis is clustered by the projected $\Delta G$ associated with a given miRNA. There doesn't seem to be a clear pattern of miRNA expression based on projected $\Delta G$.

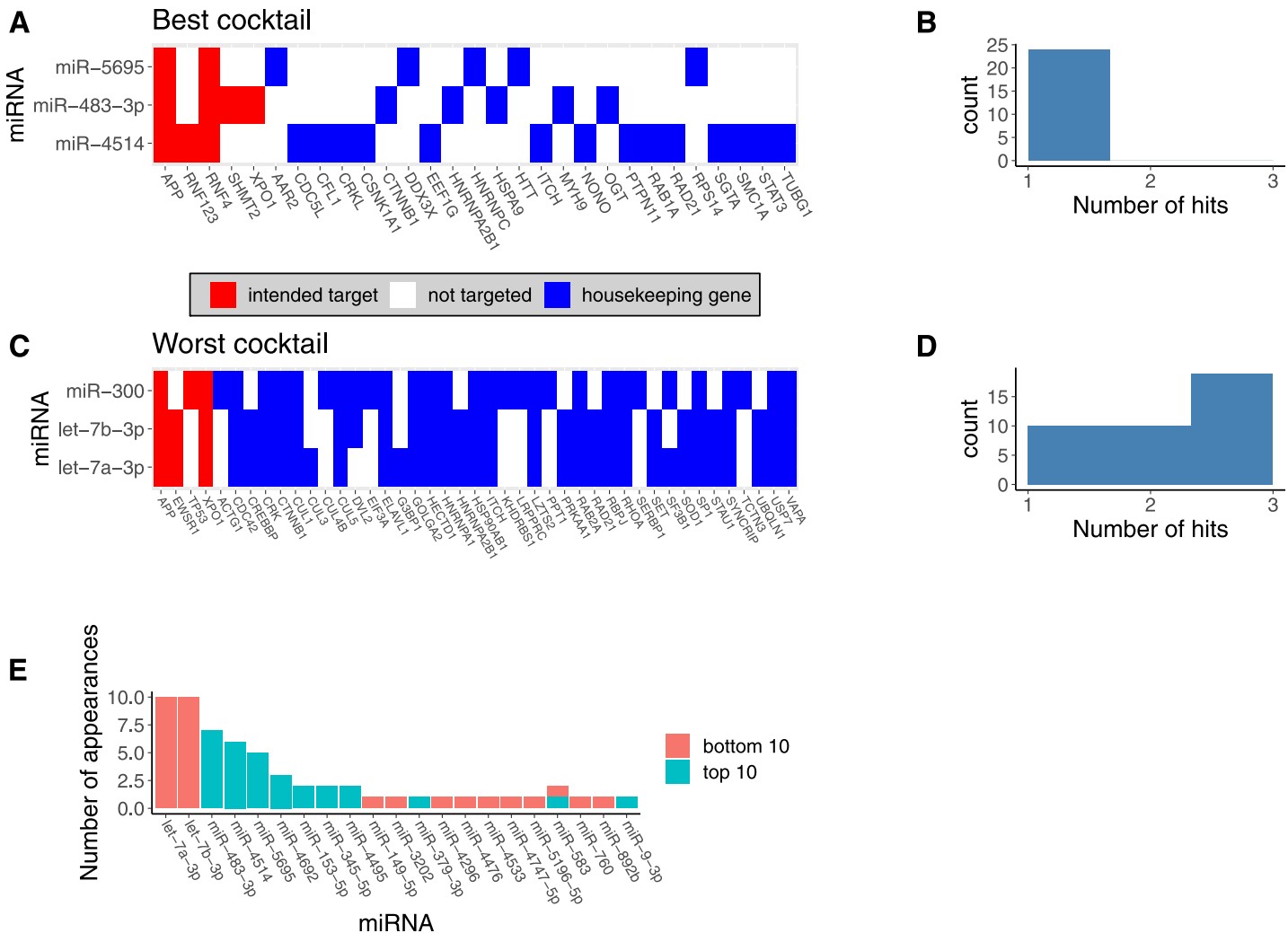

**Fig 6. We identified miR-483–3p, miR-5695, and miR-4514s as the optimal 3-miRNA cocktail for Ewing Sarcoma therapy.** We identified cocktails that are predicted to maximally downregulate target genes (red shading on the figure), while avoiding downregulation of essential housekeeping genes to limit toxicity (blue shading on the figure). **Panel A**. shows the targeting heatmap for the best predicted cocktail for cell line A673. The miRNA that make up the cocktail are presented on the y-axis. Putative gene targets are highlighted on the x-axis. Lines that span multiple miRNAs occur when a gene is downregulated by 2 or more miRNAs in the cocktail. **Panel B**. shows a histogram of the number of microRNA that target a given housekeeping gene in the best cocktail. **Panel C**. displays the targeting heatmap for the worst-performing cocktail for cell line A673 among those tested for reference. **Panel D**. shows a histogram of the number of microRNA that target a given housekeeping gene in the worst predicted cocktail. **Panel E** shows a bar graph showing the miRNA that most frequently appear in either the bottom or top 10 predicted cocktails (averaged across cell lines) for Ewing Sarcoma therapy.

because these miRNA tend to target large numbers of housekeeping genes and large numbers of genes overall. In contrast, the best miRNA cocktails tend to be composed of miRNA that target relatively few genes overall but exhibit some degree of target specificity. Put another way, they target the desired target genes while repressing relatively few essential housekeeping genes. An extreme example of this is the case of miR-483–3p. MiR-483–3p is in the bottom 50% of all miRNA when ranked by predicted network disruption, and is only predicted to repress 10 different transcripts. However, because it selectively targets several of our targets of interest, this relative small total projected network disruption is actually an attractive feature that makes it easy to build effective cocktails that include miR-483–3p. As a result, miR-483–3p appears in 7 of the top 10 predicted 3 miRNA cocktails. To assess the stability of our results, we repeated this

analysis, focusing on the top 5 or top 15 predicted protein targets. We also repeated this analysis, assuming 10% and 50% repression per miRNA that target a given mRNA. The top and bottom predicted cocktails were similar across these conditions and across all six cell lines. We have included the full ranked list of all miRNA cocktails tested across all conditions on Github.

## Discussion

In this work, we described a novel methodology for the identification of potential miRNA cocktails for Ewing sarcoma therapy. First, we performed paired miRNA and mRNA sequencing on six Ewing sarcoma cell lines (GEO accession GSE98787). We then defined a metric of cell state, network potential, based on mRNA expression and signaling network topology. Using *in silico* repression and change in network potential, we identified the most important proteins in the cell signaling network for each of the 6 cell lines. We then repeated this process for 15 patient tumor samples derived from the St. Jude Cloud [26]. Notably, this set of proteins was enriched in 24 of the 50 pathways included in the "halmarks" gene set [43, 44]. The ranked protein set was also enriched for genes involved in the canonical miRNA biogenesis pathway [6]. We then evaluated more than 16000 3-miRNA cocktails (per cell line) based on predicted ability to disrupt key proteins in the Ewing Sarcoma cell signaling network while avoiding cooperative down-regulation of essential housekeeping genes. We ranked these 3-miRNA cocktails to identify promising miRNA combinations for therapy of Ewing Sarcoma.

The protein targets and miRNA candidates we identified in our dataset are consistent with the literature on Ewing sarcoma and cancer cell signaling, suggesting biological plausibility of our methodology. Of the top 50 protein targets that we identified, 15 were previously causally implicated in cancer [41], including EWSR1, the proposed driver of Ewing sarcoma development. In addition, our network-based approach suggests that known oncogenic hub genes such as KRAS and MYC are prime targets for disruption in cancer cells. We also identified a number of plausible targets that were not previously implicated in cancer, such as MOV10. MOV10 is an RNA helicase involved in the RNA-induced silencing complex (RISC), a key complex involved in epigenetic signaling by miRNA [47]. As mentioned previously, our findings suggest that the miRNA biogenesis pathway is enriched in the setting of Ewing Sarcoma. The central role of MOV10 in the EWS cell signaling network provides further evidence for the importance of miRNA signaling in EWS oncogenesis.

Many of the miRNA we identified as potential therapeutic candidates have been previously studied due to their association with cancer outcomes, including members of the let-7 family, miR-300, miR-424–5p, miR-4282, miR-15a-5p, and miR-590–3p. Loss of expression of the let-7 family of miRNA has been widely implicated in cancer development [48–51]. In Ewing sarcoma specifically, low levels of let-7 family miRNA have been correlated with disease progression or recurrence [48]. The let-7 family of miRNA have also been studied as treatment for non-small cell lung cancer in the pre-clinical setting [15]. Loss of miR-300 has been previously correlated with development and aggressiveness of hepatocellular carcinoma [52] as well as in oncogenesis of pituitary tumors [53]. Reduced expression of miR-424–5p and miR-4282 have each been implicated in the development of basal-like breast cancer [54, 55]. MiR-15a-5p has been shown to have anti-melanoma activity [56]. In addition, miR-590–3p has been show to suppress proliferation of both breast cancer [57], and hepatocellular carcinoma [58]. The broad literature linking many of our proposed miRNA candidates for Ewing sarcoma treatment to the development and maintenance of cancer highlights the ability of our computational pipeline to identify potentially promising therapeutic candidates in this setting. Prior to application of these findings for treatment of Ewing sarcoma or any other disease, specific *in vitro* and *in vivo* validation is needed.

The process by which putative miRNA targets were selected was based on sequence homology rather than direct experimental validation. As a result, it is possible that we included false positive miRNA targets in our analysis. For this study we relied on a protein-protein interaction network presumably curated from analyzing normal human cells. It is possible that the derangements observed in cancer cells could change the underlying interaction network of a tumor cell. In the future, it may be possible to utilize protein-protein interaction networks specific to cancer or even specific to the cancer type under study. In addition, we did not consider specific binding sites that these miRNA may use to repress target mRNA. Certain miRNA may share binding sites on their target mRNA (i.e. the let-7 family of miRNA), which may make our assumption of linear additive miRNA effects invalid. We also used mRNA concentration as a surrogate for protein concentration in designing our cell signaling network. While this is not true in all cases, it is likely a reasonable approximation under steady state conditions [30–33] (see Section 0.2 for more details). In addition, protein-mRNA correlations in the cancer cell line atlas for the top proteins identified by our pipeline were fairly good, ranging from 0.07 to 0.8 for the top 50 identified protein targets. [34] (Fig 4).

Despite these limitations, our findings may facilitate the development of novel therapies for patients suffering from Ewing Sarcoma. To this point, severe toxicity has limited the translation of miRNA-based cancer therapies to the clinical setting. Our pipeline may enable the development of better miRNA therapies that clear this hurdle and open up this promising avenue of therapy for patients suffering from cancer. In addition, this novel method can facilitate the rapid identification of key proteins in any cancer cell signaling network for which mRNA sequencing data is available. This may facilitate more rapid drug discovery and assist in the discovery of proteins and miRNA that play a significant role in the cancer disease process.

## Supporting information

**S1 Fig. Network potential describes different features of a cell signaling network compared to mRNA expression alone. Panel A**: Histogram of mRNA expression for each gene (averaged across all samples). **Panel B**: Histogram of the network potential for each gene (averaged across all samples). mRNA transcripts with an expression level of zero were excluded from both histograms to better visualize the distribution of genes that are expressed. **Panel C**: Box plot showing the total mRNA expression for each cell line and patient sample (patient samples begin with SJEWS). **Panel D**: Box plot showing the total network potential for each cell line and patient sample.
(EPS)

**S2 Fig. Protein targets ranked by contribution to network stability.** When averaging across cell lines, XPO1, LMNA, EWSR1, HSP90AA1, and CUL3 were identified as the most important proteins in the Ewing sarcoma cell signaling network (when limiting our analysis to proteins causally implicated in cancer [41]). When each protein was simulated as completely repressed *in silico*, network potential was increased by 654, 456, 429, 425, and 399, respectively. The heatmap at the bottom of the plot describes the protein-mRNA correlation for each gene in the cancer cell line atlas. Grey indicates no data was available. It is reassuring that EWSR1, the kinase associated with Ewing sarcoma development, is identified as highly influential in the cell signaling network by this method.
(EPS)

**S3 Fig. Panel A**: Scatterplot with marginal histograms comparing mRNA expression to network potential. **Panel B**: Box and whisker plot showing the change in network potential for the top 50 genes, as well as 99.99% confidence intervals from the permutation test. We also

show a histogram comparing the top 50 genes identified by our pipeline using stringdb compared to biogrid as the protein-protein interaction network.
(EPS)

**S1 Table. Top protein targets for each cell line.** We ranked potential targets by predicted change in network potential when each protein was modeled as repressed.
(PDF)

**S2 Table. Top cancer-associated protein targets for each cell line.** We ranked potential targets by predicted change in network potential when each protein was modeled as repressed, limited to proteins causally associated in cancer according to the Cosmic database. Proteins that appear in the same position for $\geq 3$ cell lines are **bolded**.
(PDF)

**S3 Table. Genes ranked by network potential are enriched for several biological pathways related to cancer as well as the miRNA bio-genesis pathway.** Pathways with an adjusted p-value $< 0.05$ are shown above. "ES" refers to enrichment score and "NES" refers to the normalized enrichment score. "nMoreExtreme" refers to the number of random gene sets (out of 500) that were more enriched than the test set. Size refers to the number of genes in the pathway that were also present in our mRNA expression dataset.
(PDF)

## Acknowledgments

We acknowledge experimental support from Julia Selich-Anderson. This work made use of the High Performance Computing Resource in the Core Facility for Advanced Research Computing at Case Western Reserve University.

## Author Contributions

**Conceptualization:** Davis T. Weaver, Kathleen I. Pishas, Drew Williamson, Stephen L. Lessnick, Andrew Dhawan, Jacob G. Scott.

**Data curation:** Davis T. Weaver, Kathleen I. Pishas, Jessica Scarborough.

**Formal analysis:** Davis T. Weaver, Kathleen I. Pishas, Drew Williamson, Jessica Scarborough.

**Funding acquisition:** Kathleen I. Pishas, Stephen L. Lessnick.

**Investigation:** Davis T. Weaver, Kathleen I. Pishas, Drew Williamson, Jacob G. Scott.

**Methodology:** Davis T. Weaver, Drew Williamson, Stephen L. Lessnick, Andrew Dhawan, Jacob G. Scott.

**Resources:** Jacob G. Scott.

**Software:** Davis T. Weaver, Drew Williamson, Andrew Dhawan.

**Supervision:** Stephen L. Lessnick, Andrew Dhawan, Jacob G. Scott.

**Validation:** Davis T. Weaver.

**Visualization:** Davis T. Weaver.

**Writing – original draft:** Davis T. Weaver.

**Writing – review & editing:** Davis T. Weaver, Kathleen I. Pishas, Drew Williamson, Jessica Scarborough, Stephen L. Lessnick, Andrew Dhawan, Jacob G. Scott.

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
