## [Decision Letter · Decision Letter 0]

6 Apr 2021

Dear Dr. Scott,

Thank you very much for submitting your manuscript "Network potential identifies therapeutic miRNA cocktails in Ewings Sarcoma" for consideration at PLOS Computational Biology.

As with all papers reviewed by the journal, your manuscript was reviewed by members of the editorial board and by several independent reviewers. In light of the reviews (below this email), we would like to invite the resubmission of a significantly-revised version that takes into account the reviewers' comments. Two of the reviewers raised significant concerns regarding the procedures and the validity of the datasets used in the study. Reviewer 3 raised concerns on the premise of the study and interpretation of the results. These concerns are also shared by the Editors.

We cannot make any decision about publication until we have seen the revised manuscript and your response to the reviewers' comments. Your revised manuscript will  be sent to reviewers for further evaluation. Please note that we cannot guarantee that the revised and resubmitted manuscript will be eventually accepted for publication.

Sincerely,

Zhaolei Zhang

Associate Editor

PLOS Computational Biology

Douglas Lauffenburger

Deputy Editor

PLOS Computational Biology

Reviewer's Responses to Questions

**Comments to the Authors:**

Reviewer #1: no additional other than the uploaded document.

Reviewer #2: Weaver et al present a manuscript where network analysis is used to identify candidate microRNAs that are predicted to inhibit important genes in Ewing's sarcoma. The network analysis approach first identifies proteins (network nodes) with high potential based the mrNA expression of each node and its neighbouring nodes in the network (using a common PPI network from the literature). They then set the expression of the nodes to zero individually and measure the anticipated change in the network potential, highlighting nodes/proteins where such change is the largest. They propose the miRNA sets ('cocktails') as candidates for therapy development since these are predicted to inhibit the genes/proteins considered influential to the protein-protein network. The methodology is interesting albeit not entirely novel. Some serious limitations of molecular datasets exist, and the identified genes/proteins and targeting miRNAs could be better described in the context of biology and placement in the networks. detailed comments below.

1. the team has sequenced the transcriptomes of six Ewing's sarcoma cell lines but it is questionable whether the cell lines are representative of patient tumors vs artefacts of cell culturing. There are other RNA-seq studies representing patient tumors that could be used - https://dcc.icgc.org/releases/current/Projects/BOCA-FR;
https://www.ncbi.nlm.nih.gov/pmc/articles/PMC7191197/

2. There is no evidence or discussion about the predicted microRNAs being expressed in the tumors of interest. is there any molecular data to support it? Would the effects of predicted cocktails be attenuated if the microRNAs would be natively expressed in cells and inhibiting the predicted genes already?

3. The authors assume a linear additive effect of multiple miRNAs inhibiting a gene (line 122, page 5/16) and further, use an assumed value of 20% per microRNA. The linear, additive relationship is likely problematic if the proposes miRNAs bind similar sequences in the target mRNAs. For example, are the proposed miRNAs LET-7a-3p and LET-7b-3p commonly binding similar mRNAs and target sequences?

4. The network analysis approach would benefit from a null model that would show if the observed synthetic changes in network potential are truly significant for particular genes or could they be also observed by random chance. to achieve a null model, nodes or edges in the network could be shuffled and potential calculations of the shuffled networks would lead to a baseline distribution used as control.

5. The identified genes and microRNAs need to be better described to make the study convincing. What are the network properties of the genes/miRNAs? Are they central or hub-like? Highly or lowly expressed? known cancer driver genes? etc.

Reviewer #3: The authors describe an approach for designing therapeutic 3-microRNA (miRNA)-cocktail for treating patients with Ewing sarcoma. They first score the network potential of each mRNA based on the protein-protein interactions and their expression (Eq1). They gave high score to mRNA target based on the total network potential change after setting the mRNA target’s expression to zeros (i.e., “in-silico repression”). They then score each miRNA by total scores of its target mRNAs, which were chosen based on an ensemble of the existing sequence-based methods such as TargetScan, PicTar, Miranda, etc. The authors also made deliberate choice to avoid choosing miRNAs that target house-keeping genes to reduce toxicity.

Overall, the method is simple and easy to understand. The paper is written in organized fashion and easy to follow with a good command of language. My main concern of this paper is its actual scientific contributions. Without wet lab experiments, simulation experiments, or at least gold-standards, it is impossible to assess how the proposed 3-miRNA cocktail performs in-vitro letting alone in-vivo animal trials. There are many factors that the proposed 3-miRNA cocktail won’t work or the methodology itself is questionable.

Major comments

1. As authors pointed out in the discussion, the PPI network were not derived from cancer cells and it may very well be different from real cancer samples.

2. It is a big leap of faith that the actual patient cancer cells have the RISC machinery to effectively incorporate the 3-miRNA cocktail.

3. What’s the ideal dosage? Presumably cancer cells can increase the expression of the target genes being repressed by increasing the transcription factors’ expression, decreasing RISC proteins that are required for the miRNA repression to work, or simply increasing the protein translation

4. Repressing the genes with the largest network potentials may not necessarily improve the patient outcome. A more realistic approach would be to first look at DE genes between normal and EW patient sample (if any).

5. It’s completely unclear how to penalize miRNA scores if they do target the housekeeping genes. Currently, it seems that this step of choosing miRNA cocktail is done by mere manual inspection in post-hoc way. So the “worst-performing” or “best-performing” miRNAs are the number of house-keeping genes they target.

6. When scoring miRNAs, how to avoid redundancy when different miRNAs target the same genes in the 3-miRNA cocktails? For example, if miRNA 1 targets gene A, then the effect of miRNA 2 will not be the same as miRNA 1 if it targets the same gene. This is especially true when the abundance of miRNA 1 is higher than the abundance of gene A.

7. Why does the approach have to restrict to only the endogenous miRNAs? Can one just design exogeneous ribonucleotide oligos as in the RNA transfection to target desirable target genes?

In short, there are too many factors to be considered in order to make the claim in this paper convincing. I understand that this is a computational paper and no wet lab experiments are needed. But the topic tackled by this paper is more appropriately addressed with wet lab experiments unless gold standards or realistic simulation is carried out.

Other comments

- P5: instead of looking at house-keeping genes and then repeating the analysis without them, why not start the entire analysis without the house-keeping genes in the first place?

- Figure 3: overall the information content in this figure is low. A scatter plot may be helpful to show that the highly expressed (cancer) genes also have high network potential or maybe there is no correlation.

- P7 line 183: MiR-345-3p or 5p?

- Fig. 5. Panel B does this mean every chosen miRNA still targets 1 house keeping gene? Isn’t it still toxic?

**Have all data underlying the figures and results presented in the manuscript been provided?**

Reviewer #1: Yes

Reviewer #2: Yes

PLOS authors have the option to publish the peer review history of their article (what does this mean?). If published, this will include your full peer review and any attached files.

Reviewer #1: No

Reviewer #2: No

Reviewer #3: No

**Have the authors made all data and (if applicable) computational code underlying the findings in their manuscript fully available?**

Reviewer #3: Yes
---

## [Decision Letter · Decision Letter 1]

17 Aug 2021

Dear Dr. Scott,

Thank you very much for submitting your manuscript "Network potential identifies therapeutic miRNA cocktails in Ewings Sarcoma" for consideration at PLOS Computational Biology. As with all papers reviewed by the journal, your manuscript was reviewed by members of the editorial board and by several independent reviewers. The reviewers appreciated the attention to an important topic. Based on the reviews, we are likely to accept this manuscript for publication, providing that you modify the manuscript according to the review recommendations.

Dear authors:

We have received comments from the three reviewers. While Review 1 and 2 find the revisions satisfactory, reviewer 3 is still skeptical and pointed out several lingering issues. I ask you kindly address these concerns in the next round of the revision.

Sincerely,

Zhaolei Zhang

Associate Editor

PLOS Computational Biology

Douglas Lauffenburger

Deputy Editor

PLOS Computational Biology

[LINK]

Dear authors:

We have received comments from the three reviewers. While Review 1 and 2 find the revisions satisfactory, reviewer 3 is still skeptical and pointed out several lingering issues. I ask you kindly address these concerns in the next round of the revision.

Reviewer's Responses to Questions

**Comments to the Authors:**

Reviewer #1: I accept the paper.

Reviewer #2: The authors have addressed my comments and their revisions have improved the manuscript. I have no further comments at this point. As a minor note, figures 1-2 currently seem quite low-resolution (possibly due to the submission system).

Reviewer #3: The authors did not answer my comments in a satisfactory fashion.

For my comment 1, they said, "The use of the network we have employed is **relatively standard** in the cancer network/systems biology community." I don't think there is such *standard*. It heavily depends on how importantly the PPI network play in the analysis. In this case, the accuracy of the PPI is vital to their prediction.

Also, instead of redirecting the reviewer to a broad section like "We now more clearly note these limitations in the discussion section." to my comment 3 and other comments, the authors should consider directly write their response in the letter for the ease of reviewing. In most cases, I also don't find the relevant revised part in those sections.

Response to my comment 4 on using real EW patient data for differential analysis is simply begging the question. No effort made in seeking gold-standard or replication cohort/data.

Response to my comment 5 says "we have updated the methods (section 1.6) to reflect that our approach was in fact a priori and based on optimization over a loss function and not manual or post hoc.". I can't find the loss function. There is no equation as such.

Response to my comment 6 about additive effects is true but also highlight the flaw of the approach.

They also completely missed my 'Other comments' in my original review:

Other comments

- P5: instead of looking at house-keeping genes and then repeating the analysis without them, why not start the entire analysis without the house-keeping genes in the first place?

- Figure 3: overall the information content in this figure is low. A scatter plot may be helpful to show that the highly expressed (cancer) genes also have high network potential or maybe there is no correlation.

- P7 line 183: MiR-345-3p or 5p?

- Fig. 5. Panel B does this mean every chosen miRNA still targets 1 house keeping gene? Isn’t it still toxic?

**Have the authors made all data and (if applicable) computational code underlying the findings in their manuscript fully available?**

Reviewer #1: Yes

Reviewer #2: None

Reviewer #3: None

PLOS authors have the option to publish the peer review history of their article (what does this mean?). If published, this will include your full peer review and any attached files.

Reviewer #1: No

Reviewer #2: No

Reviewer #3: No

Figure Files:

Data Requirements:

Reproducibility:

References:

---

## [Editor Report · Decision Letter 2]

20 Sep 2021

Dear Dr. Scott,

We are pleased to inform you that your manuscript 'Network potential identifies therapeutic miRNA cocktails in Ewing sarcoma' has been provisionally accepted for publication in PLOS Computational Biology.

Best regards,

Zhaolei Zhang

Associate Editor

PLOS Computational Biology

Douglas Lauffenburger

Deputy Editor

PLOS Computational Biology

---

## [Editor Report · Acceptance letter]

13 Oct 2021

PCOMPBIOL-D-21-00209R2 

Network potential identifies therapeutic miRNA cocktails in Ewing sarcoma

Dear Dr Scott,

I am pleased to inform you that your manuscript has been formally accepted for publication in PLOS Computational Biology. Your manuscript is now with our production department and you will be notified of the publication date in due course.

With kind regards,

Andrea Szabo
